# On the Use of Weighted Least-Squares Approaches for Differential Interferometric SAR Analyses: The Weighted Adaptive Variable-lEngth (WAVE) Technique

**DOI:** 10.3390/s20041103

**Published:** 2020-02-18

**Authors:** Francesco Falabella, Carmine Serio, Giovanni Zeni, Antonio Pepe

**Affiliations:** 1School of Engineering, The University of Basilicata, 85100 Potenza, Italy; francesco.falabella@unibas.it (F.F.); carmine.serio@unibas.it (C.S.); 2National Research Council of Italy, Institute for the Electromagnetic Sensing of the Environment (CNR-IREA), 80124 Napoli, Italy; zeni.g@irea.cnr.it; 3National Research Council of Italy, Institute of Methodologies for Environmental Analysis (CNR-IMAA), Tito Scalo, 85050 Potenza, Italy

**Keywords:** ground displacement, synthetic aperture radar, small baseline, variable-length time-series, weighted least-squares

## Abstract

This paper concentrates on the study of the Weighted Least-squares (WLS) approaches for the generation of ground displacement time-series through Differential Interferometric SAR (DInSAR) methods. Usually, within the DInSAR framework, the Weighted Least-squares (WLS) techniques have principally been applied for improving the performance of the phase unwrapping operations as well as for better conveying the inversion of sequences of unwrapped interferograms to generate ground displacement maps. In both cases, the identification of low-coherent areas, where the standard deviation of the phase is high, is requested. In this paper, a WLS method that extends the usability of the Multi-Temporal InSAR (MT-InSAR) Small Baseline Subset (SBAS) algorithm in regions with medium-to-low coherence is presented. In particular, the proposed method relies on the adaptive selection and exploitation, pixel-by-pixel, of the medium-to-high coherent interferograms, only, so as to discard the noisy phase measurements. The selected interferometric phase values are then inverted by solving a WLS optimization problem. Noteworthy, the adopted, pixel-dependent selection of the “good” interferograms to be inverted may lead the available SAR data to be grouped into several disjointed subsets, which are then connected, exploiting the Weighted Singular Value Decomposition (WSVD) method. However, in some critical noisy regions, it may also happen that discarding of the incoherent interferograms may lead to rejecting some SAR acquisitions from the generated ground displacement time-series, at the cost of the reduced temporal sampling of the data measurements. Thus, variable-length ground displacement time-series are generated. The mathematical framework of the developed technique, which is named Weighted Adaptive Variable-lEngth (WAVE), is detailed in the manuscript. The presented experiments have been carried out by applying the WAVE technique to a SAR dataset acquired by the COSMO-SkyMed (CSK) sensors over the Basilicata region, Southern Italy. A cross-comparison analysis between the conventional and the WAVE method has also been provided.

## 1. Introduction

Weighted Least-squares (WLS) is a generalization of ordinary least-squares (LS) in which non-negative weights are attached to data points to take into account the varying quality of the available data while solving systems of linear equations [1]. Ordinary and weighted least-squares optimization problems typically arise when the differential synthetic aperture radar interferometry (DInSAR) technique is employed for the generation of displacement maps of the Earth’s surface [2,3,4]. The DInSAR technique is based on the measurement of the phase difference (i.e., interferogram) between two synthetic aperture radar (SAR) images acquired at different times and from slightly diverse orbital positions. As a result, the measurement of the projection of the ground displacement along the radar sensor line-of-sight (LOS) is provided [5,6]. Temporal and spatial decorrelation artefacts [5,6,7] are responsible for additive noise in the generated interferograms. Specifically, long interferometric time separations and large spatial baselines (i.e., the separation between the orbital positions) lead to decorrelated interferograms. However, temporal and spatial decorrelation effects are distinct for different interferograms for every SAR pixel. The use of weights in the various existing DInSAR tools aims to take into account the distinctive stochastic properties of the interferograms. Although the DInSAR technology was initially established to investigate single episodes of ground displacement [8,9], it is nowadays widely adopted, mainly, for the extraction and the analysis of the temporal evolution of the ground deformation through advanced Multi-Temporal interferometric SAR (MT-InSAR) techniques [10,11,12,13,14,15,16,17,18,19,20,21,22,23,24,25,26,27,28,29]. In the last 20 years, several MT-InSAR algorithms have progressively been developed to study the classes of natural (e.g., [30,31,32,33]) and human-induced (e.g., [34,35,36,37,38]) ground deformation phenomena. The MT-InSAR techniques can usually be grouped into the two families of the Persistent Scatterers [10,11,12,13] and the Small Baseline [14,15,16,17,18,19,20] approaches, even though some hybrid PS/SB techniques have also been developed (e.g., [21]). In particular, the PS algorithms work using all single-master interferometric SAR data pairs, i.e., computed with respect to one single SAR scene (the master image). They allow estimating the surface changes of very stable, point-wise scatterers that retrain their coherence over time. In this case, there is no need to impose any restrictions on the temporal and spatial separation (i.e., the baseline) of the used InSAR data pairs. Conversely, the SB methods are predominantly concentrated on detecting and monitoring the ground displacement of scatterers that are distributed over several resolution cells of the focused images. Such distributed scatterers (DS) are, however, more prone to be corrupted by spatial and temporal decorrelation phenomena [5,6,7]. Accordingly, to diminish the noise in the generated interferograms, only multiple-master InSAR data pairs characterized by small baselines (temporal and spatial) are pre-selected. The corresponding differential SAR interferograms are then computed and properly inverted to obtain information on the temporal evolution of the Earth’s surface displacement. The SB MT-InSAR techniques [14,15,16,17,18] rely on the solution of systems of linear equations using ordinary LS or L-1 norm minimization (e.g., [39]). Among the SB methods, the Small Baseline Subset (SBAS) technique [14] has been the pioneering one. It permits analysing the deformation of DS objects on the ground by inverting a sequence of SB interferograms via an LS optimization. Specifically, if the selection of the SB interferograms creates disconnected sets of data separated by large baselines, the constraint on the minimum norm velocity of the ground displacement time-series is imposed, and the Singular Value Decomposition (SVD) method [40] is applied. Finally, in the framework of the methods addressing the DS targets signal characterization and study, the SqueeSAR technique [23] and other alternative multi-temporal approaches have also been recently proposed [24,25,26,27]. Globally, the performance of all the developed MT-InSAR methods depends on the accurateness of the phase unwrapping and phase inversion operations. Both these key operations are more robust against noise artefacts if the phase stability of every single, pixel-based measurement is taken into account. In this framework, the use of weights may guarantee improved performances in terms of error noise propagation and ground displacement measurements accuracy. The phase unwrapping (PhU) operation [41,42,43,44,45,46,47,48,49,50,51,52,53,54,55] is the crucial step in any InSAR processing tool. It involves the searching of the (unknown) 2π-integer multiples that must be added to the wrapped phases to compute the unwrapped (full) phases. Historically, PhU algorithms have been developed to process single interferograms independently and have worked both on regular (e.g., [41]) and irregular (e.g., [50]) grids of pixels. Regular PhU procedures assume that all phase measurements are reliable. On the contrary, on irregular grids, the PhU operations are only performed on a group of SAR pixels with medium-to-high-coherence, by first computing from them a planar-connected graph (e.g., a Delaunay triangulation [56]). The low coherent pixels are discarded from the subsequent analyses, or their relevant phase information is recovered by spatial interpolation of neighbouring coherent SAR pixels. Over the arcs of the computed graph, the vectors of the wrapped phase differences are subsequently calculated. The unwrapped phases of the pixels over the graph are obtained as the solution of a weighted minimum L-p norm problem, where the weights are set by considering the phase quality across the given spatial arcs. With *p* = 2, PhU is seen as an LS optimization problem, whose analytic solutions are known [43,53,54,55]. In this context, the weights are employed to constrain the unwrapped phases being less dependent on the phase integration paths across the noisy regions. As in [57], the phase variances of the interferograms are exploited to set the weights used for achieving the WLS solution. Several weighted LS PhU techniques are currently available in the literature (e.g. [53,54,55]). However, the L-2 norm tends to smooth discontinuities, thus not preserving the 2π-difference restriction between the unwrapped and wrapped phases. For this reason, the L-1 norm is commonly preferred. The L-1 norm PhU problem has been resolved by Flynn [47] and Costantini [49,50], by reformulating it as the solution of a Minimum Cost Flow (MCF) network optimization. With the advent of the MT-InSAR era, the interest has progressively moved towards the development of three-dimensional PhU algorithms that could allow the generation of a stack of multi-temporal unwrapped maps, taking profit from the knowledge of the spatial and temporal relationships among the computed interferograms. Some 3-D [58,59] and hybrid space-time [60,61] PhU algorithms have been developed in the last 20 years. Weighted Least Squares (WLS) approaches have also been used for inverting sequences of differential SAR interferograms and producing ground deformation products, where they are used to improve monitoring performance of ground surface motions, see for instance [62,63,64,65].

In this paper, we propose a novel WLS method that naturally extends the usability of the SBAS technique [14] in regions with medium-to-low coherence. Specifically, the process is based on the adaptive selection, pixel-by-pixel, of a proper set of phase measurements coming from coherent interferograms, which are then inverted using a WLS approach. As a result, for the different analysed SAR pixels of the imaged scene, different interferometric SAR distributions are used, selected taking into account the stability of the involved phase signals. In particular, the weights are set for every SAR pixel as the reciprocal of the InSAR phase variance, computed by considering the stochastic properties of the SAR interferograms. As a consequence of the adaptive selection procedure, it may also happen that SAR data can be grouped into several disjointed subsets (as in the original SBAS procedure [14]), which are straightforwardly linked by using a weighted generalization of the singular value decomposition (WSVD) method [66,67,68]. In some very critical noisy regions, it may also happen that the adaptive (pixel-by-pixel) selection of the “good” interferograms could lead to the removal of some SAR images. In these cases, the displacement time-series can still be generated, but at the expense of the temporal sampling of data measurements, thus having the so-called variable-length deformation time-series. Although the displacement time-series of some noisy SAR pixels have a reduced length, the temporal sampling of the obtained deformation time-series is still suitable to extract useful information on the ground motion at the corresponding spatial locations. As a consequence, the number of analysable SAR pixels significantly increases. Our method, which is named Weighted Adaptive Variable-lEngth (WAVE), shares some similarities with the intermittent SBAS (ISBAS) technique [69,70]. The ISBAS technique estimates the mean velocity deformation rate as well as the DEM errors related to the intermittent coherent targets, and subsequently applies the conventional, un-weighted SBAS inversion strategy to the residual interferograms, obtained after compensating the estimated components. The WAVE method exacerbates the original idea at the base of the ISBAS technique, allowing for the generation of the variable-length LOS-projected ground displacement time-series related to the intermittent coherent pixels. As said, this is done by combining the adaptive selection of the coherent pixels with a weighted singular value decomposition (WSVD) inversion strategy, directly applied to the whole (i.e., the coherent and intermittent coherent) set of SAR pixels. The idea of using different networks of interferograms, depending on the phase quality of the measurements, can also further be extended and applied in the general framework of the SB DInSAR methods, beyond the SBAS procedure. The experimental results have been performed considering a dataset composed of 50 SAR images, gathered by the Italian Space Agency COSMO-SkyMed (CSK) constellation sensors over the Basilicata region, Southern Italy.

## 2. Weighted Least-Squares Inversion of SB Differential SAR Interferograms

Let us assume the availability of a group of N+1 SAR images acquired over the same region of Earth at the ordered times $\left[t_{0},t_{1},\ldots ,t_{N}\right]$. Every SAR image is preliminarily co-registered to the same SAR scene and, starting from them, a group of M interferometric SAR data pairs is selected. Although the proposed method can also potentially work with large baseline interferograms, only small baseline InSAR data pairs are profitably chosen to reduce the overall algorithm computational burden. To this aim, the temporal and spatial baselines of the interferograms are taken into account for the identification of a suitable set of small baseline (SB) InSAR data pairs. The stack of differential SAR interferograms, namely $\Delta \Phi \left(P\right)=\left[\Delta \phi_{0}\left(P\right),\Delta \phi_{1}\left(P\right),\ldots ,\Delta \phi_{M-1}\left(P\right)\right]$, where P is a generic radar pixel of the azimuth/range {*Az* × *Rg*} spatial domain, is then generated. This operation is done by simulating and subtracting the relevant topographic phase components from the InSAR phase differences, computed using the orbital information and a digital elevation model (DEM) of the area [71]. 

### 2.1. Summary of the Un-Weighted SBAS Inversion Technique

The SBAS procedure starts by assuming the availability of a sequence of M unwrapped interferograms, namely $\Delta \Psi \left(P\right)=\left[\Delta \psi_{0}\left(P\right),\Delta \psi_{1}\left(P\right),\ldots ,\Delta \psi_{M-1}\left(P\right)\right]$, computed in correspondence with the generic radar pixel P of the azimuth/range spatial domain. The unwrapped phases can be obtained using either conventional 2-D [41,42,43,44,45,46,47,48,49,50,51,52,53,54,55], 3-D [58,59], or hybrid space-time [60,61] PhU algorithms. In particular, in this work, we apply a two-step PhU processing chain based on an adaptation of the EMCF algorithm [60], as discussed in Section 2.2. The unwrapped interferograms are expressed as follows:(1)$$\begin{array}{l} \Delta \psi_{k}\left(P\right)=\frac{4\pi }{\lambda }d_{k}\left(P\right)+\frac{4\pi }{\lambda }\cdot \frac{b_{⊥k}}{r\left(P\right)\sin\vartheta \left(P\right)}\cdot \Delta z\left(P\right)+\varepsilon_{k}\left(P\right),\;\;\;\;\\ k=0,1,\ldots ,M-1;\;\;\;\;\forall P\in \left\{Az\times Rg\right\} \end{array}$$
where $\frac{4\pi }{\lambda }d_{k}\left(\cdot \right),\;\;k=0,1,2,\ldots ,M-1$ is the phase term that accounts for the relative radar LOS-projected deformation (which also includes the contributions due to the inhomogeneity of the atmosphere) of the *k-th* interferogram, where λ is the operational wavelength, and $b_{⊥k}\left(\cdot \right),\;\;k=0,1,2,\ldots ,M-1$ is the perpendicular baseline of the *k-th* InSAR data pair. Moreover, Δz(·) represents the height profile error, and r(·) and ϑ(·) are the slant-range distance and the radar side-looking angle, respectively. Furthermore, the noise phase term $\varepsilon_{k}\left(\cdot \right),\;\;k=0,1,2,\ldots ,M-1$ accounts for phase unwrapping errors, incorrect orbital information, and decorrelation noise artefacts. 

The system of linear Equation (1) defines an LS optimization problem, which is solved as fully detailed in [14]. Precisely, for every identified pixel of the SAR image scene, the topographical error term Δz(·) as well as a rough estimate of the deformation rate, namely x(·), are first estimated [14,72]. Subsequently, the unwrapped phases are corrected from the topographical errors, thus obtaining:(2)$$\begin{array}{l} \Delta {\tilde{\psi }}_{k}\left(P\right)=\Delta \psi_{k}\left(P\right)-\frac{4\pi }{\lambda }\cdot \frac{b_{⊥k}}{r\left(P\right)\sin\vartheta \left(P\right)}\cdot \Delta z\left(P\right)=\frac{4\pi }{\lambda }d_{k}\left(P\right)+\varepsilon_{k}\left(P\right),\;\;\;\;\\ k=0,1,\ldots ,M-1;\;\;\;\;\forall P\in \left\{Az\times Rg\right\} \end{array}$$

The system of linear Equation (2) can be synthetically expressed as [14]:(3)$$\Delta \tilde{\Psi }=A\cdot \Phi +E$$
where **A** is the M×N incidence-like matrix of the linear transformation of Equations (2) and (3), $\Phi =\left(\frac{4\pi }{\lambda }\right)D$ is the vector of the (unknown) deformation values $D\left(P\right)=\left[0,d_{1}\left(P\right),\ldots ,d_{N}\left(P\right)\right]$ at the given analysed SAR pixel (where it is implicitly assumed that $d_{0}\left(P\right)\equiv 0$), and **E** is the vector of the error noise contributions. System (3) is solved in the LS sense. Nevertheless, in the more general case that the selected SB InSAR data pairs are arranged to form some independent subsets of data that are separated by large baselines [14], the matrix **A** turns out to be rank-deficient. In this case, the problem becomes:(4)$$\begin{array}{c} \Phi :min{||A\cdot \Phi -\Delta \tilde{\Psi }||}_{2}\\ min{||\Phi ||}_{2} \end{array}$$

Problem (4) can be solved by decomposing the matrix **A** using the singular value decomposition method [67,68,69]. However, a solution that minimizes the norm of Φ is not physically feasible because it is responsible for significant discontinuities in the final deformation result. This problem was circumvented in [14] by manipulating the system of Equations (3) and replacing the phase unknowns $\Phi \left(P\right)=\frac{4\pi }{\lambda }D=\left[0,\phi_{1}\left(P\right),\ldots ,\phi_{N}\left(P\right)\right]$ with the phase velocities between time-adjacent acquisitions $V\left(P\right)=\left[v_{1}\left(P\right),\ldots ,v_{N}\left(P\right)\right]$, which are defined as $v_{k}\left(\cdot \right)=\left(\phi_{k}\left(\cdot \right)-\phi_{k-1}\left(\cdot \right)\right)/\left(t_{k}-t_{k-1}\right),k=1,2,\ldots ,N$.

As a result, Problem (4) becomes:(5)Φ:min||B·V−ΔΨ˜||2min||V||2
which is solved using the SVD method applied to the matrix  B(M≥N) (see [14] for its mathematical expression). Mathematically, the matrix **B** is decomposed as follows:(6)B=Q·S·ZT
where **Q** is an (M×M) orthogonal matrix, **Z** is an orthogonal (N×N) matrix, and **S** is a diagonal (M×N) matrix, whose diagonal elements represent the singular values [σ1,σ2,…,σN] of the matrix **B**. Accordingly, the solution of Problem (5) is obtained as:(7)V=B+·ΔΨ˜
with B+ being the inverse generalized of B:B+=Z·S+·QT, and S+ is the diagonal matrix S+=diag[1σ1,1σ2,…,1σN,0,…,0]∈ℝN×M. The phase values associated with every SAR acquisition are finally obtained by time-integrating the estimated values of the vector V. Once collected, the phases associated with the SAR acquisitions are converted to deformation measurements, and subsequently the Atmospheric Phase Screen (APS) is computed. Finally, the APS-filtered deformation time-series [10,14] are calculated.

### 2.2. EMCF-Based Multi-Temporal Phase Unwrapping

In this sub-section, we present an adaptation of the Extended Minimum Cost Flow (EMCF) PhU algorithm [60]. First, we give some introductory information on the EMCF phase unwrapping algorithm. The EMCF technique [60] represents an extension of the minimum cost flow PhU method [49,50]. It exploits two triangulations; the former is computed in the Temporal/Perpendicular baseline domain. Indeed, every SAR image can be represented in that domain with a point, and the arcs connecting different points represent the selected interferometric SAR data pairs. Those arcs are organized to form a (reduced) triangulation in the Temporal/Perpendicular baseline plane, see Figure 1, obtained by first computing a triangulation starting from the available SAR data and then discarding those arcs that involve at least one large baseline interferogram [60]. The latter triangulation is computed in the {*Az* × *Rg*} plane and encompasses the group of coherent SAR pixels that are common to the entire pool of the selected SB interferograms. The unwrapping operation is then performed. First, we identify all the arcs of the computed “spatial” triangles, and, for every arc, we independently carry out a “temporal” PhU step, which is done by applying the MCF technique [49,50] to the temporal/perpendicular baseline grid. The second stage of the EMCF PhU procedure uses, for every arc, the previously computed unwrapped phases as the starting point for the successive spatial phase unwrapping operations. The latter are performed on every single interferogram by using the conventional MCF technique [49,50].

As initially explored in [72], here, we propose to adapt the EMCF procedure to enlarge the pool of the interferograms to be unwrapped and improve the overall PhU performance. To this aim, the selected M interferometric SB SAR data pairs are grouped into two different sets, labelled as $\left\{S_{1}\right\}$ and $\left\{S_{2}\right\}$, which are composed of $M_{1}=M^{\left(Tr\right)}$ and $M_{2}=M-M_{1}$ interferograms, respectively. The interferograms related to the first set identify the primary network, which forms a (reduced) triangulation in the temporal/perpendicular baseline plane (see Figure 2a) consisting of $M^{\left(Tr\right)}$ data pairs, see [58] for further details, also potentially selected using the sub-optimal procedure described in [22]. 

The primary network is ideally suited to the application of the original EMCF PhU technique. Let ${\left[\Delta \phi_{0}^{\left(1\right)}\left(P\right),\Delta \phi_{1}^{\left(1\right)}\left(P\right),\ldots ,\Delta \phi_{M^{\left(Tr\right)}-1}^{\left(1\right)}\left(P\right)\right]}^{T}$ be the set of wrapped phases and ${\left[\Delta \psi_{0}^{\left(1\right)}\left(P\right),\Delta \psi_{1}^{\left(1\right)}\left(P\right),\ldots ,\Delta \psi_{M^{\left(Tr\right)}-1}^{\left(1\right)}\left(P\right)\right]}^{T}$ be the corresponding set of the unwrapped phases, where *P* is a generic SAR pixel of the {*Az* × *Rg*} spatial grid. The PhU operations are performed over a set of highly-coherent SAR pixels that are in common to the whole set of interferograms of $\left\{S_{1}\right\}$, namely $G^{\left(Tr\right)}\equiv \left\{P\in S_{1}\right\}$. These pixels can be identified either by computing the average spatial coherence of the interferograms or exploiting the phase triangulation closures (see the appendix of [73]). 

For the subsequent operations, the identification of the coherent SAR pixels that are specific for every SAR interferogram represents a very critical step. Over the interferograms of the primary network, as earlier said, a group of high-coherence SAR pixels is selected, and the unwrapping operations are done only on those SAR pixels. After, considering only the interferograms related to the primary network, a rough estimate of the LOS-projected displacement time-series, namely ${\hat{d}}^{\left(1\right)}\left(P\right)=\left[0,{\hat{d}}_{1}^{\left(1\right)}\left(P\right),{\hat{d}}_{2}^{\left(1\right)}\left(P\right),\ldots ,{\hat{d}}_{N^{\prime }}^{\left(1\right)}\left(P\right)\right],\;\;\forall P\in G^{\left(Tr\right)}$ and the DEM errors $\Delta \hat{z}\left(P\right),\;\;\forall P\in G^{\left(Tr\right)}$ are obtained by applying the (conventional) un-weighted SBAS procedure (see Section 2.1) or alternative methods. As a result of the removal of some triangles (i.e., those that involve at least one large-baseline interferogram) from the complete triangulation in the temporal/perpendicular baseline plane [60], it is possible that the reduced triangulation may involve $N^{\prime }+1<N+1$ distinctive SAR images. However, this limitation of the original EMCF PhU technique is straightaway circumvented by complementing the interferograms of the primary network (see Figure 2a) with some additional interferograms (see Figure 2b). 

The effective unwrapping procedure of the additional SAR interferograms, namely ${\left[\Delta \phi_{0}^{\left(2\right)}\left(P\right),\Delta \phi_{1}^{\left(2\right)}\left(P\right),\ldots ,\Delta \phi_{M_{2}-1}^{\left(2\right)}\left(P\right)\right]}^{T}$, which belong to the second set $\left\{S_{2}\right\}$, is hereinafter detailed. These added interferograms are independently unwrapped following a strategy that is similar to the one initially adopted in [72]. In particular, for every interferometric SAR data pair, the phase unwrapping operation is performed over the relevant set of coherent SAR pixels (which are selected by imposing a threshold on the corresponding spatial coherence map) by using the conventional MCF procedure [50]. First, for every SAR pixel, we consider the local topography errors $\Delta \hat{z}\left(P\right),\;\;\;\;\forall P\in G^{\left(Tr\right)}$ and the deformation time-series ${\hat{d}}^{\left(1\right)}\left(P\right)=\left[0,{\hat{d}}_{1}^{\left(1\right)}\left(P\right),{\hat{d}}_{2}^{\left(1\right)}\left(P\right),\ldots ,{\hat{d}}_{N^{\prime }}^{\left(1\right)}\left(P\right)\right],\;\;\;\;\forall P\in G^{\left(Tr\right)}$ (including possible atmospheric artefacts) obtained by processing the sequence of the primary network interferograms. The collected rough LOS-projected displacement time series, computed on the unwrapped interferograms of the first network, are temporally low-pass filtered and time-interpolated, and subsequently spatially interpolated to retrieve information on the incoherent pixels. The temporal and spatial interpolation steps are both performed using a simple polynomial interpolator. Thus, the estimated displacement time-series and the residual topographic components are used to generate a model of the (unknown) unwrapped phases. In particular, by considering the *k-th* interferogram of the group of the added interferograms $\left\{S_{2}\right\}$, the phase model has the following expression:(8)$$m_{k}=\frac{4\pi }{\lambda }\left\{\Delta {\hat{d}}_{k}^{\left(1\right)}+\frac{b_{⊥k}^{\left(2\right)}}{r\sin\vartheta }\Delta \hat{z}\right\},\;\;\;\;k=0,1,\ldots ,M_{2}-1$$
where $\Delta {\hat{d}}_{k}^{\left(1\right)}$ is the LOS-projected relative deformation of the *k-th* InSAR data pair, $\Delta \hat{z}$ is the DEM error, and $b_{⊥k}^{\left(2\right)}$ is the perpendicular baseline of the *k-th* InSAR data pair. The modelled phase contributions are then modulo-2π subtracted from the phase of the interferograms, and the residual phase is then unwrapped by the conventional MCF technique [50]. 

Hence, the whole set of the unwrapped interferograms is obtained. We note that for every SAR interferogram of the primary network, the previously collected unwrapped phases are spatially interpolated over the specific set of coherent pixels of the given InSAR data pair, identified by considering those pixels with a spatial coherence value larger than 0.2.

The diagram flow of the whole process leading to the unwrapping of the complete set of the interferograms is shown in Figure 3.

### 2.3. Weighted Adaptive Variable-lEngth (WAVE) InSAR Method 

In this sub-section, we will introduce the adaptation of the methods presented in Section 2.1 to the weighted case. The presented analysis is performed adaptively for every pixel P of the {*Az* × *Rg*} domain. As a starting point, let us consider the whole sequence of the unwrapped interferograms, say ${\left[\Delta \psi_{0}\left(P\right),\Delta \psi_{1}\left(P\right),\ldots ,\Delta \psi_{M-1}\left(P\right)\right]}^{T}$, again. As earlier stated, for a given SAR pixel P, only a subset of the available M interferograms has an adequate spatial coherence value (i.e., larger than a prescribed threshold, for instance $\hat{\gamma }=0.2$). Differently from [14] (and similarly to [70]), we propose to discard the interferograms that are less coherent (adaptively on a pixel-by-pixel basis). More precisely, for a given SAR pixel *P*, we propose to discard all the interferograms exhibiting a spatial coherence value less than $\hat{\gamma }=0.2$ and use the remaining ones for the generation of ground displacement time-series. Accordingly, for every single SAR pixel of the imaged scene, an exclusive network of DInSAR interferograms is selected and, then, inverted to generate the LOS-projected displacement time-series relevant to the selected SAR pixel. As an example, Figure 4 shows two different networks of InSAR data pairs related to two distinctive SAR pixels of the analysed area. More precisely, the DInSAR distributions in Figure 4a,b are composed by 242 and 77 interferometric data pairs, respectively. 

Let us now describe the subsequent inversion step of the unwrapped interferograms. Our procedure has some similarities with [63,69] but, in our case, the selection and removal of some interferometric SAR data pairs may lead, as in [14], to an undetermined problem and (potentially) to the generation of variable-length ground displacement time-series. 

Before going in detail, let us first spend a few additional moments on the setting of the weights associated with the phase measurements, in correspondence with the SAR pixels that have a coherence value larger than the considered threshold (e.g., in our experiments, $\hat{\gamma }=0.2$). To this aim, the variance of the multi-looked interferometric phases is, first, estimated pixel-by-pixel. It is well known that the density probability function (pdf) of the wrapped multi-looked phase $\phi^{\left(ML\right)}$ is [7,74,75]:(9)$$\begin{array}{cc} p_{\phi^{\left(ML\right)}}\left(\phi^{\left(ML\right)}\right)= & \frac{\Gamma \left(L+0.5\right){\left(1-\gamma^{2}\right)}^{L}\;\gamma \cos\left(\phi^{\left(ML\right)}-{\bar{\phi }}^{\left(ML\right)}\right)}{2\sqrt{\pi }\Gamma \left(L\right){\left(1-\gamma^{2}\cos^{2}\left(\phi^{\left(ML\right)}-{\bar{\phi }}^{\left(ML\right)}\right)\right)}^{L+0.5}}\\ & +\frac{{\left(1-\gamma^{2}\right)}^{L}}{2\pi }F\left(L,1;0.5;\right)\gamma^{2}\cos^{2}\left(\phi^{\left(ML\right)}-{\bar{\phi }}^{\left(ML\right)}\right) \end{array}$$
where L is the number of the looks, ${\bar{\phi }}^{\left(ML\right)}$ is the average multi-looked (true) phase, Γ(·) is the gamma function, and F(·) is the hypergeometric function. Accordingly, the phase variance can be numerically calculated as:(10)$$\sigma_{\phi^{\left(ML\right)}}^{2}=\overset{+\pi }{\underset{-\pi }{\int }}{\left[\phi^{\left(ML\right)}-{\bar{\phi }}^{\left(ML\right)}\right]}^{2}p_{\phi^{\left(ML\right)}}\left(\phi^{\left(ML\right)}\right)d\phi^{\left(ML\right)}$$

However, for the low coherence regions, the estimated coherence values typically result in being biased. Moreover, the phase variance calculation is somewhat computationally expensive, requiring the numerical integration in Equation (10). Alternatively, we can exploit the basic principles of the directional statistics [76]. Indeed, under some simplified assumptions [76], it can be shown that a suitable approximation for the pdf of the wrapped phase (i.e., a directional data vector) is the wrapped normal distribution WN(ϑ;μ,ρ), which is obtained by wrapping onto the circle the normal distribution $N\left(\mu ,\sigma^{2}\right)$, that is:(11)$$WN\left(\vartheta ;\mu ,\rho \right)=\frac{1}{\sigma \sqrt{2\pi }}\overset{+\infty }{\underset{q=-\infty }{\sum }}\exp\left\{\frac{-{\left(\vartheta -\mu +2q\pi \right)}^{2}}{2\sigma^{2}}\right\}$$
where $\rho =\exp\left[-\frac{\sigma^{2}}{2}\right]$ is the resultant mean length of the circular phase data [76].

In our case, if we refer to the *k-th* SAR differential interferogram and the generic (multi-looked) SAR pixel P, the circular data represents the unitary phase vector $\left[\phi_{k,0},\phi_{k,1},\ldots ,\phi_{k,L-1}\right]$. It is relevant to the single-look phases that averaged together (see Figure 5), one gets the *k-th* multi-looked phase at the SAR pixel P*,* where L is the number of averaged phase samples. Accordingly, [76]:(12)$$\phi_{k}^{\left(ML\right)}\left(P\right)=\arctan\frac{\sum_{h=0}^{L-1}\sin\phi_{k,h}}{\sum_{h=0}^{L-1}\cos\phi_{k,h}};\;\;\;\;\rho =\frac{1}{L}\left|\sum_{h=0}^{L-1}\exp\left[j\phi_{k,h}\right]\right|$$
where $j=\sqrt{-1}$ is the imaginary unit, and:(13)$$\sigma_{\phi_{k}^{\left(ML\right)}}^{2}=-2\ln\rho =-2\ln\left\{\frac{1}{L}\left|\sum_{h=0}^{L-1}\exp\left[j\phi_{k,h}\right]\right|\right\}$$

Hence, the phase variances can be easily computed by Equation (13), and the DInSAR interferograms are generated without requiring highly demanding computations as with Equation (10). Alternatively, it is possible to consider the Cramer-Rao bound for the phase variance [6], which is expressed as follows:(14)$$\sigma_{\phi_{k}^{\left(ML\right)}}^{2}=\frac{1-\gamma^{2}}{2L\gamma^{2}}$$

Once the phase variances are estimated, a suitable way to set the weights that are associated with every single InSAR measurement (i.e., the multi-looked interferometric phase related to every InSAR data pair) is to consider the reciprocal of the phase variances [29], namely $W=\left[\omega_{0},\omega_{1},\ldots ,\omega_{M-1}\right]$ with: $\omega_{i}=1/\sigma_{\phi_{i}^{\left(ML\right)}}^{2}\;\;\;i=0,1,\ldots ,M-1$. Using the matrix formalism, we can also write that:(15)$$\left\{diag\left[\omega_{0},\omega_{1},\ldots ,\omega_{M-1}\right]\right\}={\left\{diag\left[\sigma_{{\Delta \Psi }_{0}}^{2},\sigma_{{\Delta \Psi }_{1}}^{2},\ldots ,\sigma_{{\Delta \Psi }_{M-1}}^{2}\right]\right\}}^{-1}$$

Equation (15) is fundamental for the analysis of the error propagation in the weighted case. To introduce the problem, we refer to the fundamental principles of the uncertainty theory applied to the linear regression problems [77]. We note that, as earlier described, the un-weighted SBAS procedure leads to the solution of the optimization Problem (4), which is done by Equation (7). For a given SAR pixel, the known term $\Delta \tilde{\Psi }$, representing the vector of the unwrapped phases, is a random phase vector, and $C_{\Delta \tilde{\Psi }}$ is its related covariance matrix:(16)$$C_{\Delta \tilde{\Psi }}=\left[\begin{array}{cccc} \sigma_{0,0}^{2} & \sigma_{0,1}^{2} & \ldots & \sigma_{0,M-1}^{2}\\ \sigma_{1,0}^{2} & \sigma_{1,1}^{2} & \ldots & \sigma_{1,M-1}^{2}\\ \ldots & \ldots & \ldots & \ldots \\ \sigma_{M-1,0}^{2} & \sigma_{M-1,1}^{2} & \ldots & \sigma_{M-1,M-1}^{2} \end{array}\right]$$
where the elements on the principal diagonal of the covariance matrix represent the phase variances of the M DInSAR interferograms used for the SBAS inversion, and the off-diagonal element of $C_{\Delta \tilde{\Psi }}$, say $\sigma_{i,j}^{2}\;\;\;\;i\neq j$, represents the covariance between the (unwrapped) phases of the *i-th* and the *j-th* InSAR data pair. By applying the basic principles of the linear regression and the uncertainty theory [77] to Equation (7), the covariance matrix of the unknown velocity vector V is derived as follows: (17)$$C_{V}=B^{+}\cdot C_{\Delta \tilde{\Psi }}\cdot B^{+T}$$

In our treatment, although the computed interferograms have a given grade of dependence (because obtained by the same sequence of SAR data), we assume that the interferometric phases are uncorrelated. This simplified assumption is widely adopted in the literature, and only very few InSAR studies have addressed the problem of the systematic calculation of the correlation among a group of InSAR interferograms computed from the same set of SAR data [78]. Under this simplified hypothesis, Equation (17) becomes: (18)$$C_{V}=B^{+}\cdot diag\left\{\sigma_{0,0}^{2},\sigma_{1,1}^{2},\ldots ,\sigma_{M-1,M-1}^{2}\right\}\cdot B^{+T}$$

Let us now describe the developed Weighted Adaptive Variable-lEngth (WAVE) DInSAR technique, which permits us to select and adaptively process a set of differential SAR interferograms. Let us refer, again, to a generic SAR pixel  P of the multi-looked grid, and let ${\left[\Delta \psi_{0},\Delta \psi_{1},\ldots ,\Delta \psi_{M-1}\right]}^{T}$ be the corresponding vector of the unwrapped phases, ${\left[\gamma_{0},\gamma_{1},\ldots ,\gamma_{M-1}\right]}^{T}$ be the relevant spatial coherences, and $W=\left[\omega_{0},\omega_{1},\ldots ,\omega_{M-1}\right]$ be the weights. As earlier said, for every SAR pixel, a preliminary selection of the “good” interferograms is made up. Hence, only the group of coherent interferograms survives. Therefore, let $I=\left[i_{0},i_{1},\ldots ,i_{\hat{M}-1}\right]$ be the indexes of the coherent interferograms, with $\hat{M}$ being the number of selected ones. The extracted vectors of the unwrapped phases and the relevant weights are $\Delta \hat{\Psi }={\left[\Delta {\hat{\psi }}_{0},\Delta {\hat{\psi }}_{1},\ldots ,\Delta {\hat{\psi }}_{\hat{M}-1}\right]}^{T}={\left[\Delta \psi_{i_{0}},\Delta \psi_{i_{1}},\ldots ,\Delta \psi_{i_{\hat{M}-1}}\right]}^{T}$ and ${\left[\omega_{i_{0}},\omega_{i_{1}},\ldots ,\omega_{i_{\hat{M}-1}}\right]}^{T}$ respectively. Similarly to [14], because of the adaptive selection of the interferograms to be used and the discarding of the remaining ones, the SAR data can generally be organized in disjointed subsets, which have to be properly linked. In our approach, we also check the temporal overlapping of the produced data subsets. From our analyses, we discard those SAR pixels for which there is no time overlap between the data subsets. Note also that the extracted interferograms now do not necessarily involve all the available SAR images. Indeed, only a group of SAR acquisitions are preserved, let ${\left[{\hat{\phi }}_{0},{\hat{\phi }}_{1},\ldots ,{\hat{\phi }}_{\hat{N}}\right]}^{T}$ be the unknown phase relevant to those $\hat{N}+1$ “survived” SAR images, which are those acquired at ordered times ${\left[{\hat{t}}_{0},{\hat{t}}_{1},\ldots ,{\hat{t}}_{\hat{N}}\right]}^{T}$. As for the un-weighted SBAS case, the problem is still formulated concerning the vector of the unknown velocities between contiguous SAR images $\hat{V}={\left[v_{1},v_{2},\ldots ,v_{\hat{N}}\right]}^{T}$, where ${\hat{v}}_{k}\left(\cdot \right)=\left({\hat{\phi }}_{k}-{\hat{\phi }}_{k-1}\right)/\left({\hat{t}}_{k}-{\hat{t}}_{k-1}\right)\;\;\;\;k=1,2,\ldots ,\hat{N}$. 

The weighted inversion problem to be solved is: (19)$$\begin{array}{l} min{||B\cdot \hat{V}-\Delta \hat{\Psi }||}_{W}\\ \;min{||\hat{V}||}_{2} \end{array}$$
where the matrix B is also adaptively obtained for the given SAR pixel taking into account the distribution of the “survived” SAR acquisitions and the relevant InSAR data pairs, following the same lines of [14]. The norm indicated in Equation (19) is defined as in [79], namely ||X||W=(XT·W·X)12. Accordingly, Problem (19) leads to the minimization of the following norm:(20)||B·V^−ΔΨ^||W=[(B·V^−ΔΨ^)T·W·(B·V^−ΔΨ^)]12==ω0ε0+ω1ε1+⋯+ωM^−1εM^−1
subjected to a minimum-norm constraint on the velocity. Note that, in our case, $W=diag\left(\omega_{0},\omega_{1},\ldots ,\omega_{\hat{M}-1}\right)$, see Equation (15), and the vector e represents the phase residuals between the unwrapped phases and those reconstructed from the retrieved solution, namely $e=B\cdot \hat{V}-\Delta \hat{\Psi }$. 

The Problem (20) represents the particularization of the following more general weighted least-squares problem:(21)min||B·V^−ΔΨ^||Wmin||V^||T
where the matrixes W (ℝM^×M^) and T (ℝN^×N^) put weights on the row and column of the matrix B (ℝM^×N^), respectively; its solution is unique. It is obtained, see [79], by first calculating the W**-**T**-**singular decomposition of matrix B [79], assuming that M^≥N^, which decomposes B such that U−1·B·Z=diag(μ1,μ2,…,μN^), where μi,    i=1,…,N^ are the singular values of the decomposition, U is orthogonal to W**,** and Z is orthogonal to T. 

Note that, the W**-**T decomposition of matrix B is obtained [79] as follows. First, the matrixes of the constraints W and T are written by the Cholesky factorization as: $W=L\cdot L^{T},\;\;T=K\cdot K^{T}$. Then, the SVD decomposition of the matrix $C=L^{T}\cdot B\cdot K^{-T}$ is obtained, such as $Q^{T}\cdot C\cdot X=diag\left(\mu_{1},\mu_{2},\ldots ,\mu_{\hat{N}}\right)$. Finally, the decomposition of the matrix B is done by setting $U=L^{-T}\cdot Q$ and $Z=K^{-T}\cdot X$.

The final solution is eventually obtained as follows:(22)V^=Z·DB+·U−1·ΔΨ^
where: Z∈ℝN^×N^, DB+=[1/μ1,1/μ2,…,1/μr,0,0,..,0]∈ℝN^×M^ with r≤N^, and U∈ℝM^×M^.

In our case, matrix T is the identify matrix of ℝN^×N^ and, by Equation (5), the weights are set as W=diag{[1/σΔΨ^02,1/σΔΨ^12,…,1/σΔΨ^M−12]}. Accordingly, the matrixes W and T are diagonal and: L=W1/2,    T=K=IN^×N^,    C=W1/2·B. As a consequence, we have:(23)V^=X·DB+·Q−1·W1/2·ΔΨ^=Ω·W1/2·ΔΨ^

Of course, the solution (23) is now pixel-dependent, because both the selected InSAR data pairs and the associated weights are different from pixel to pixel. In critical regions, it may also happen that the pre-selection of the “good” interferograms, for the given SAR pixel, leads some SAR acquisitions to be discarded. In this latter case, the problem in (19)–(21) is mathematically defined in the same way as in the previous sub-section. Still, the vectors of the unknown velocities have a reduced length (i.e., $\hat{N}$ is less than N). As a consequence, once the estimated velocity values are time-integrated to recover the corresponding LOS-projected deformation time-series, namely $\hat{\phi }=\left[{\hat{\phi }}_{0},{\hat{\phi }}_{1},\ldots ,{\hat{\phi }}_{\hat{N}-1}\right]$, the latter have a reduced length (i.e., less than N+1).

The diagram flow chart of the WAVE method is shown in Figure 6. 

We would like to remark that although the proposed WAVE method has been developed in the framework of the SB approaches, its main characteristics allow a simple extension to the more general non-SB context.

Finally, let us spend a few moments on the propagation of errors for the weighted case. By considering Equation (22), it is easy to prove that:(24)$$C_{\hat{V}}={\hat{B}}^{+}\cdot C_{\Delta \hat{\Psi }}\cdot {\hat{B}}^{+T}$$
where ${\hat{B}}^{+}=\Omega \cdot W^{1/2}$, and $C_{\Delta \hat{\Psi }}$ is the covariance matrix of the selected unwrapped phases. Under the hypothesis that the interferograms are uncorrelated, namely $C_{\Delta \hat{\Psi }}=W^{-1}$, Equation (24) simplifies as:(25)$$C_{\hat{V}}=\Omega \cdot W^{1/2}\cdot W^{-1}\cdot W^{1/2}\cdot \Omega^{T}=\Omega \cdot \Omega^{T}$$

This outcome is significant: It means that if the weights are chosen as $\left\{diag\left[\omega_{0},\omega_{1},\ldots ,\omega_{M-1}\right]\right\}={\left\{diag\left[\sigma_{{\Delta \Psi }_{0}}^{2},\sigma_{{\Delta \Psi }_{1}}^{2},\ldots ,\sigma_{{\Delta \Psi }_{M-1}}^{2}\right]\right\}}^{-1}$, the co-variance matrix of the velocity terms is constant for every element of the (estimated) velocity vector, thus implying that errors propagate in the computed deformation time-series uniformly over time.

To check the quality of the obtained variable-length LOS displacement time-series, a weighted version of the temporal coherence factor [60], which also takes into account the weights ${\left[\omega_{i_{0}},\omega_{i_{1}},\ldots ,\omega_{i_{\hat{M}-1}}\right]}^{T}$ of the selected interferograms, has also been introduced:(26)$$\Gamma \left(Q\right)=\frac{1}{\sum_{k=0}^{\hat{M}\left(Q\right)-1}\omega_{k}}\left|\overset{\hat{M}\left(Q\right)-1}{\underset{k=0}{\sum }}\omega_{k}\exp\left\{j\left[\Delta {\hat{\psi }}_{k}-{\hat{\phi }}_{IM_{k}}+{\hat{\phi }}_{IS_{k}}\right]\right\}\right|$$
where $\hat{M}\left(Q\right)$ is the number of above-coherence-threshold interferograms at the given pixel Q, and $IM_{k}$ and $IS_{k}$ are the indexes of the master and slave time acquisitions of the *k-th* InSAR data pair. 

## 3. Dataset and Study Area

The study area where we have successfully applied the WAVE methodology encompasses the Basilicata region, Southern Italy (see Figure 7). The territory of Basilicata is predominantly mountainous. It is surrounded by the Mediterranean Sea, with a relatively small flatland zone near the coastal area overlooking the Ionian Sea, which is one of the two outlets to the sea, where the other ones are on the Tyrrhenian side. Moreover, the region is crossed by the Apennine Mountains that extend from the Ligurian Alps to the Northern Sicily mounts, with famous massifs, such as Mount Sirino (2005 m), Mount Pollino (2267 m), Mount Volturino (1835 m), and many other ones. The hills are almost entirely composed of clay soil, which is subjected to erosion processes caused by the frequent rainfalls, mostly in the winter season. These processes are the starting trigger for landslides, which have afflicted and still afflict several cities [80,81,82,83,84], such as Montemurro, Latronico, Pomarico, Maratea, Craco, and Montescaglioso. It is also important to underline the high seismic risk of the entire area, as confirmed by the numerous earthquakes [85,86,87] that have occurred during the last 200 years, which, in many cases, have destroyed several small cities [88], where thousands of people tragically died. The intertwining of these issues, including also the hydrogeological instability, contributes to the overall instability of zone. That results in a problematic scenario for correctly applying the conventional interferometric SAR techniques. The available SAR dataset is composed of 50 SAR acquisitions, collected by the COSMO-SkyMed (CSK) sensors of Italian Space Agency (ASI), from which a set of 263 differential SAR interferograms were generated. 

The distribution of the SAR images in the temporal/perpendicular baseline plane, as well as the identified set of the SB InSAR data pairs, are shown in Figure 1. COSMO-SkyMed [89] is a constellation of four SAR sensors, both operating at the centre carrier frequency of 9.6 GHz with a maximum radar bandwidth of 400 MHz, single and dual-polarization, which flight at an altitude of about 620 Km from the sea level with an inclination angle of 97.86°. From the point of images acquisition, it is possible to exploit three types of scene shooting, the Spotlight-2 mode with the best resolution achievable (1 m) and a 10 Km × 10 Km swath, the StripMap (HIMAGE and PING-PONG) mode with a medium resolution (3 m and 15 m) and a medium swath area (40 Km and 30 Km), and the ScanSAR (WIDE and HUGE REGION) mode with the lower resolution (16 m and 30 m) and the greatest swath possible (100 Km and 200 Km) with these sensors. 

For our purpose, we have used HH polarized StripMap (HIMAGE) images, collected from February 2012 to November 2018 along ascending orbit, each with a right-looking radar direction and viewing angles of approximately 34°. Every raw SAR acquisition has been first focused [90,91] to obtain the focused In-Phase and Quadrature terms for each acquisition, e.g., the Single Look Complex (SLC) images. The information of the entire SAR dataset is summarized in Table 1, where for every SAR image, we list: (i) the acquisition dates, (ii) the (average) Doppler Centroid (DC) and (iii) the orthogonal baseline (calculated w.r.t the common, reference master image acquired on 1 October 2014). Every SAR image covers a small area of the Basilicata region, between the meridians ranging from 15.6° to 16.0° and the parallels ranging from 40.0° to 40.4°. In this zone, the Pertusillo dam, the Lagonegrese, and the Val D’Agri territory, as well as the Vallo di Diano territory, are included. In Figure 8, an optical image of the area is shown, and some locations of particular interest are highlighted.

## 4. Experimental Results 

In this section, we show the results achieved by applying the WAVE method to the set of SAR data presented in Section 3. The experiments were conducted by first selecting from the available *N =* 50 SAR acquisitions a group of InSAR data pairs, selected by imposing a maximum perpendicular separation of 800 m and a maximum time span of 2 years, respectively. As a result, we have identified *M =* 263 InSAR data pairs. Once chosen, the *M* differential SAR interferograms have been generated. To this purpose, precise satellite orbital information and a three-arc Shuttle Radar Topography Mission (SRTM) digital elevation model (DEM) of the area have been used to remove the topographic phase components. The computed differential SAR interferograms have also been independently multi-looked (10 azimuth and 10 range looks, respectively) and pre-filtered by using the approach in [92]. As explained in Section 2, the interferometric SAR data pairs are organized into two independent groups. The former corresponds to the primary interferometric data pairs network shown in Figure 2a, and the latter represents the set of additional interferograms (see Figure 2b). These two sets of interferograms are then unwrapped by using the strategy outlined in Section 2.2.

To investigate the potential of the WAVE approach, we first processed the sequence of the *M* unwrapped interferogram using the conventional, un-weighted SBAS approach. Figure 9 shows the mean displacement velocity map of the investigated area, overlaid to an amplitude SAR image of the zone. Only the SAR pixels with a temporal coherence greater than 0.6 have been depicted on the map. We also show in Figure 9b–e the LOS-projected displacement time-series of some selected SAR pixels, which are located in areas subjected to sensible deformation signals. The shown displacement time-series are obtained after the estimation and removal of the Atmospheric Phase Screen (APS). The latter operation is carried out by the cascade of a High-Pass temporal filter and a Low-Pass spatial filter [10] applied to the achieved LOS-projected, un-filtered deformation time-series. 

Subsequently, we have applied the WAVE method to the same set of unwrapped interferograms. We remark that, even though we pre-selected a set of SB interferograms, all the possible DInSAR interferograms could be potentially processed. However, this would turn out to be extremely time-consuming. Similarly to [14], however, the removal of the low-coherent interferograms may lead to the generation of multiple disjointed subsets, which are linked by using the weighted version of the SVD strategy, as detailed in Section 2. Furthermore, in some noisy areas, the possibility of having LOS-projected displacement time-series with variable length also arises. The application of the WAVE technique to the whole set of SAR pixels has led to the generation of the corresponding LOS-projected displacement time-series. To check the quality of the achieved results, we have calculated the weighted temporal coherence; see Equation (26), namely ${\left\{\Gamma \left(P\right)\right\}}_{P\in \left\{Az\times Rg\right\}}$. With respect to the temporal coherence factor initially introduced in [60], in our case, we only consider the set of DInSAR interferograms that has actually been used during the inversion step for the generation of the surface displacement time-series. Accordingly, the weighted temporal coherence factor turns out to be more sensitive to the unwrapping errors in the medium-to-high coherent regions. Conversely, the weighted temporal coherence does not take into account the decorrelation artefacts that were present in the interferograms that were adaptively discarded, because of the low values of their spatial coherence. Additional constraints have also been imposed to discard some extra SAR pixels with low quality. To do that, we have generated three spatial maps containing, for every SAR pixel, the information related to: (i) the number of interferograms actually used during the inversion of the unwrapped phases, namely ${\left\{N_{I}\left(P\right)\right\}}_{P\in \left\{Az\times Rg\right\}}$; (ii) the number of time acquisitions of the computed variable-length displacement time-series, namely ${\left\{N_{D}\left(P\right)\right\}}_{P\in \left\{Az\times Rg\right\}}$; and (iii) the number of independent data subsets that are linked by the WSVD decomposition described in Section 2.3. Figure 10 shows the map of the number of used interferograms, as obtained in our case. Finally, the set of the well-processed SAR pixels are those that satisfy the following multiple conditions:(27)$$\left\{\Gamma >\hat{\gamma }\right\}\cap \left\{N_{I}>\iota \right\}\cap \left\{N_{D}>\delta \right\}\cap \left\{N_{I}\geq N_{D}\right\}$$
where $\hat{\gamma }$,  ι and  δ are the thresholds on the weighted temporal coherence, the number of used interferograms, and the number of used data, respectively. 

Figure 11 shows the map of the mean displacement rate of the area as obtained by using the WAVE approach, where only the well-processed SAR pixels satisfying the multiple Conditions (27) have been portrayed. As a result, we got a total number of 696,898 well-detected SAR pixels against the 132,399 corresponding values obtained by processing the same set of unwrapped interferograms using the un-weighting SBAS procedure, with an improvement of roughly 526%. We also show in Figure 11b–e four examples of the obtained LOS-projected displacement time-series. In particular, Figure 11b,c show the plots related to two SAR pixels (see their location in the map of Figure 11a) characterized by one single data subset and with full-length displacement time-series (i.e., consisting of 50 time samples), and Figure 11d,e show the plots related to two SAR pixels with variable-length deformation time-series. We also show in Figure 12a–d the relevant networks of InSAR interferograms, as represented in the temporal/perpendicular baseline plane, and related to the SAR pixels whose LOS displacement time-series are shown in Figure 11b–e.

Finally, the LOS-projected displacement time-series have been compared to available external GPS measurements. GPS data were downloaded from the website [93] and projected along the sensor line-of-sight. Figure 13 shows the results of this cross-comparison analysis in correspondence to the seven GPS stations available in the area. The red stars in Figure 13 highlight the location of the GPS sites in Figure. Table 2 shows the values of the root mean square error (RMSE) of the difference between the obtained LOS-projected displacement time-series and the corresponding GPS measurements.

## 5. Discussion

We now discuss the main topics that emerged in this work, taking into account the attained experimental results. We have developed a hybrid InSAR method that complements the adaptive selection of a suitable set of interferograms with a weighted least squares inversion strategy, to improve our capability to monitor the evolution of surface displacement, especially in the low-coherent regions, by considerably increasing the number of measurements points. This is done, in correspondence with some SAR pixels, at partial expenses of the time sampling of the measurements (i.e., the number of sampling SAR acquisition times). Therefore, we have produced the so-called variable-length LOS-projected displacement time-series. The benefit of this approach with respect to the conventional, un-weighted SBAS procedure is that it preserves all the principal features of the SBAS technique (and potentially of other alternative SB approaches) but complements it with a spatial adaptive selection procedure, which allows working always with moderate-to-high coherent interferograms, thus mitigating the impact of noise decorrelation and phase unwrapping artefacts. Even though the WAVE method represents an extension of the SBAS procedure, it does not impose, in principle, any pre-selection of the small baseline InSAR data pairs to be used. For reducing the computational burden of the algorithm, not all possible InSAR interferograms are generated, but only a fraction of them. Because of this, the WAVE approach does not necessarily belong to the class of the SB methods. Even in the large baseline interferograms, some isolated pixels still preserve their coherence, and they can be profitably used for the generation of ground displacement time-series. Therefore, our approach avoids the time-consuming inspection (and in some cases, the potential manual removal) of the InSAR pairs to be used for the generation of the deformation time-series. The use of the multiple-criterion (27) for the identification of the well-processed SAR pixels also allows quantifying the improvement in terms of analysable SAR pixels. The weights used during the interferograms inversion procedure are essential to reduce the impact of noise artefacts in the obtained DInSAR products and, as demonstrated by Equation (25), they are set in such a way to have LOS-projected displacement time-series with uniformly time-distributed uncertainty values. The WAVE method can be applied to any set of multi-looked SAR interferograms, no matter the InSAR tools or the pre-processing (e.g., the noise-filtering) methods that have been used to generate and process the interferograms. In addition, the WAVE method can be straightforward implemented in concurrent and/or parallel computing architectures, thus only requiring independent, pixel-by-pixel analyses. Further efforts are required to study how to set these weights in the most general case that the interferograms are correlated (as they are). Also, the WAVE method has to be validated in wider geophysical contexts, especially in areas with moderate-to-low coherence, where the un-weighted SBAS approach (and other alternative SB methods) still suffers from significant decorrelation noise and phase unwrapping disturbances. All these issues are matters for future investigations.

## 6. Conclusions

This study has addressed the problem of generating variable-length LOS-projected displacement time-series in moderate-to-low coherent areas using a hybrid InSAR method, which is named as Weighted Adaptive Variable-lEngth (WAVE) technique. The paper provides an introduction of the weighted least-squares (WLS) InSAR methods, with a specific focus on those applied in the small baseline InSAR framework. The mathematical and statistical fundamentals of the WAVE method have been detailed in the work. The pros of the WAVE method are essentially related to the possibility to extend the number of detectable pixels, but at the potential costs of the measurement time sampling. However, this represents a reasonable compromise, especially in moderate-to-high-vegetated areas, such as for the Basilicata region, where the decorrelation noise and the phase unwrapping mistakes may lead to unsatisfactory results. The experimental results have shown the validity of the proposed method in the selected study area. The new WAVE method requires further extensive studies to demonstrate its general applicability for the retrieval of deformation signals in various geophysical contexts. As earlier stressed, we would like to remark that the WAVE technique naturally fits with the requirements of the SBAS technique as well. 

## Figures and Tables

**Figure 1 sensors-20-01103-f001:**
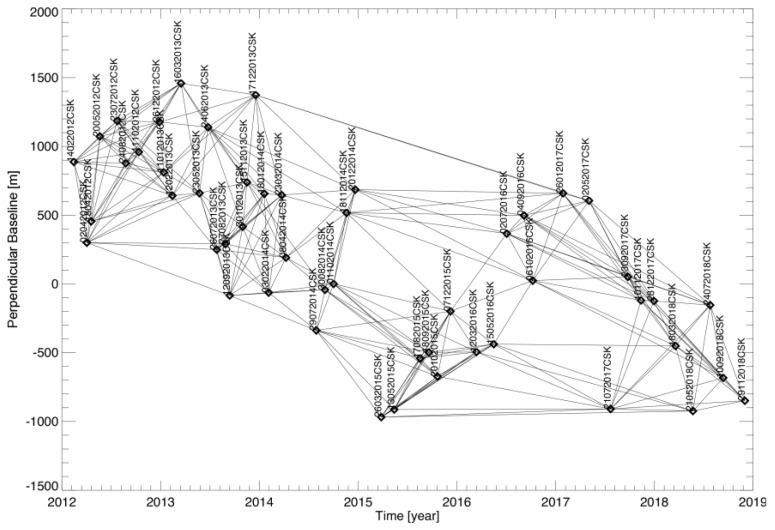
Distribution of the SAR data into the temporal/perpendicular baseline plane for the COSMO-SkyMed dataset of the Basilicata region, used for the experiments shown in Section 4. In particular, the dataset is composed by 50 SAR images. Starting from these data, M = 263 small baseline InSAR pairs are identified. The dates are represented in the “day-month-year” format, and the label “CSK” stands for the satellite mission.

**Figure 2 sensors-20-01103-f002:**
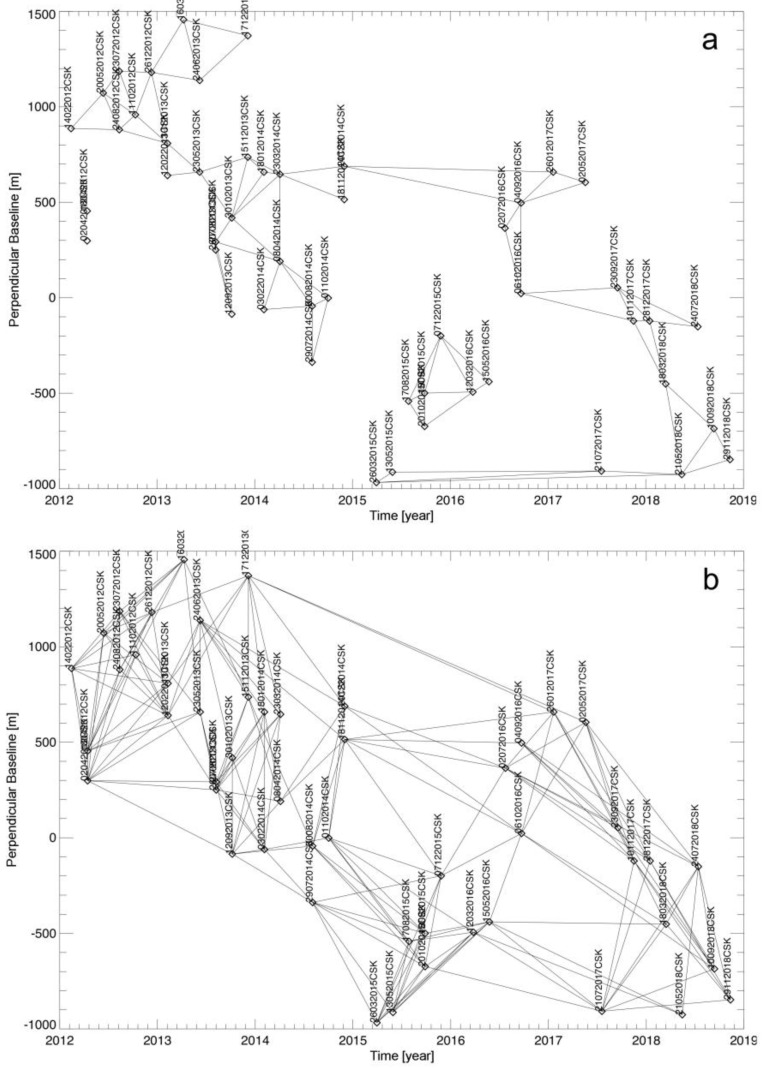
Interferometric SAR distributions in the temporal/perpendicular baseline plane. (**a**) The $M_{1}=M^{\left(Tr\right)}$ interferometric data pairs of the (reduced) Delaunay triangulation used by the canonical EMCF PhU algorithm. (**b**) The additional $M_{2}=M-M_{1}$ interferometric data pairs that are unwrapped using the modified EMCF PhU algorithm.

**Figure 3 sensors-20-01103-f003:**
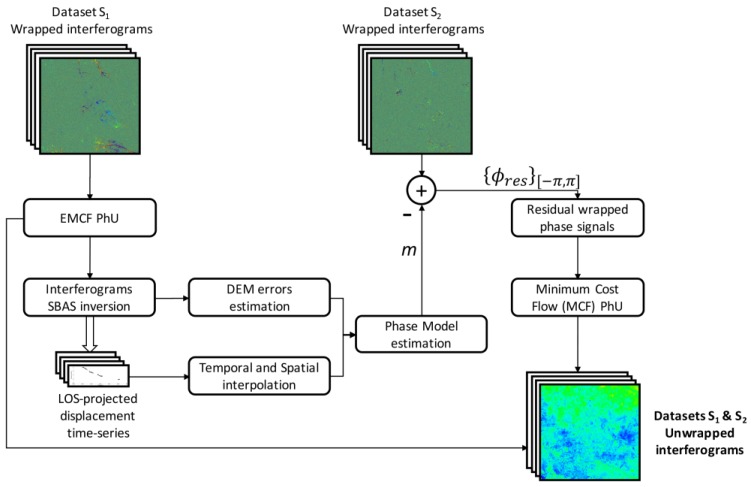
Flow Chart of the modified EMCF Phase-Unwrapping (PhU) procedure.

**Figure 4 sensors-20-01103-f004:**
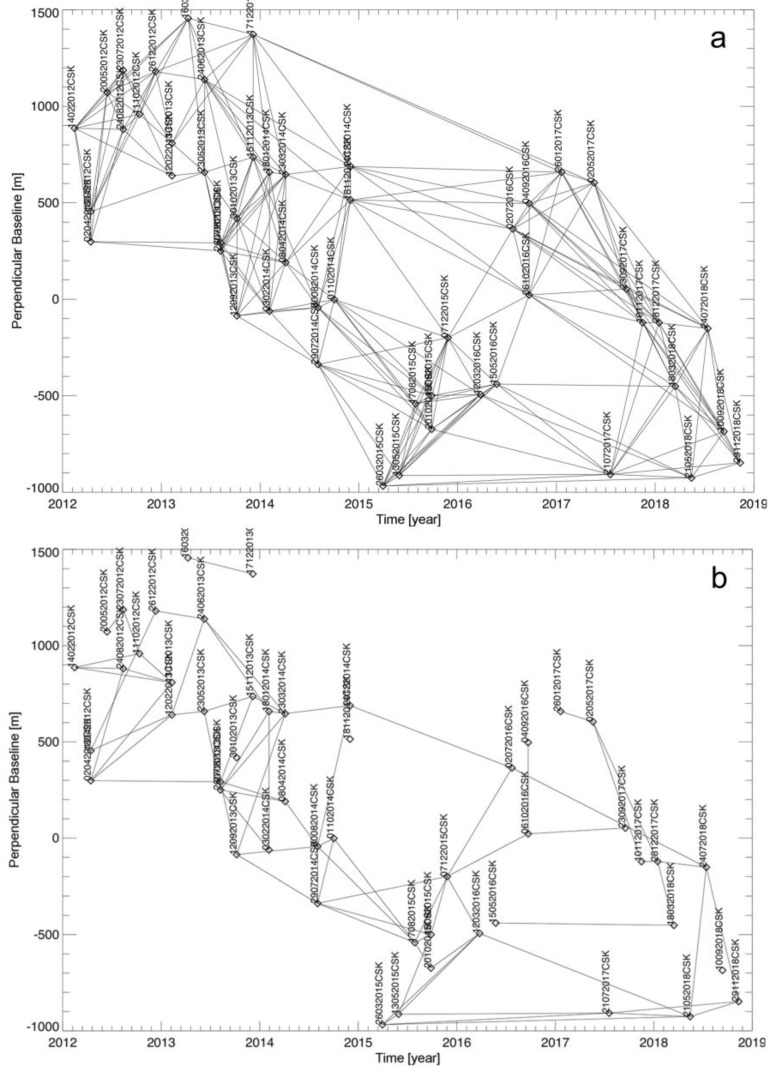
The adaptive SAR distributions in the temporal/perpendicular baseline planes relative to two given SAR pixels. The adaptive network in (**a**) consists of 242 InSAR data pairs, whereas that shown in (**b**) consists of 77 InSAR data pairs.

**Figure 5 sensors-20-01103-f005:**
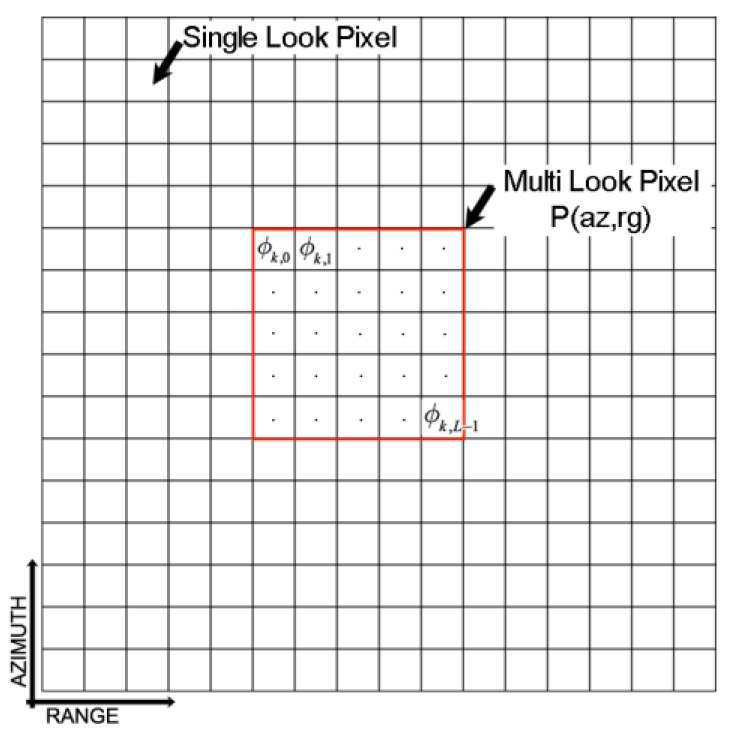
Sketch of the pixel grid of the *k-th* interferogram exploited for spatial multi-look operations. In the red box are highlighted the single-look phases used for the computation of the variance relative to the generic multi-looked pixel *P* of radar coordinates (az,rg).

**Figure 6 sensors-20-01103-f006:**
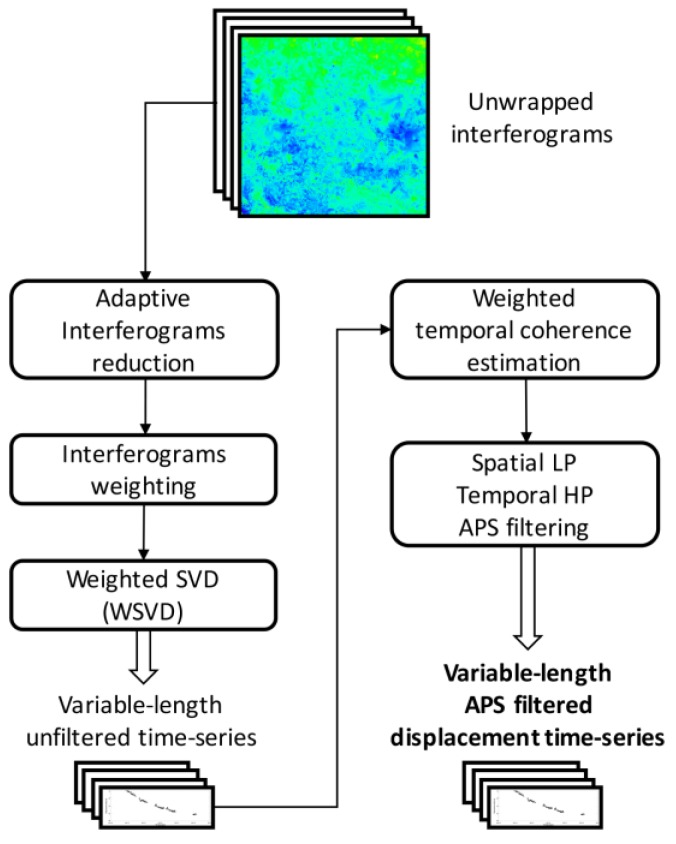
Flow chart of the WAVE algorithm.

**Figure 7 sensors-20-01103-f007:**
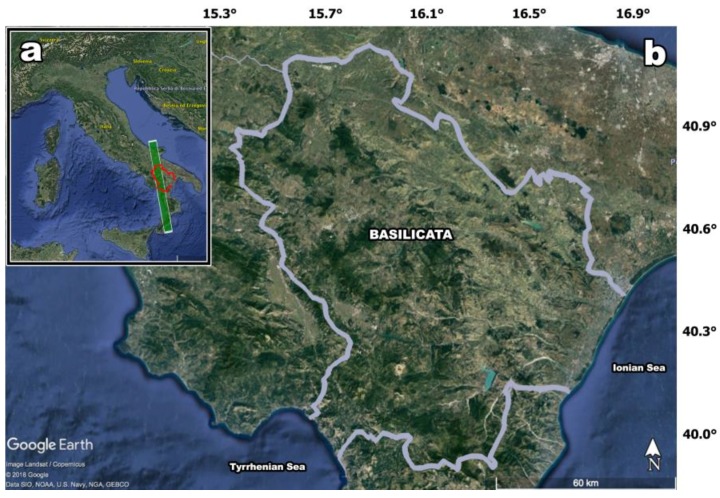
Geography of the study area: (**a**) The Italian Peninsula. The red demarcation lines identify the Basilicata region; the green rectangle identifies the used COSMO-SkyMed ascending strips. (**b**) Macro of the Basilicata region, where it is possible to see the two sea outlets. Shown are the Google Earth’s optical images acquired by Landsat satellites on 14 December 2015.

**Figure 8 sensors-20-01103-f008:**
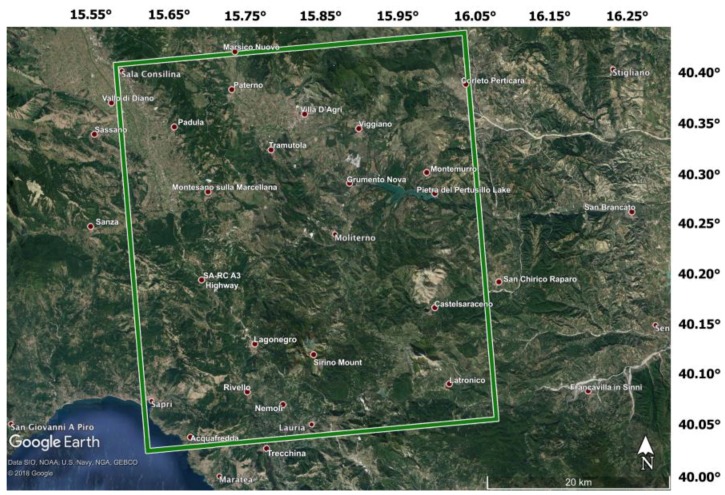
Optical image of the study area, covered by the COSMO-SkyMed acquisitions used for the tests. The footprint of the processed CSK frame is highlighted by a green rectangle. The Google Earth optical images have been acquired by the Landsat satellites on 14 December 2015.

**Figure 9 sensors-20-01103-f009:**
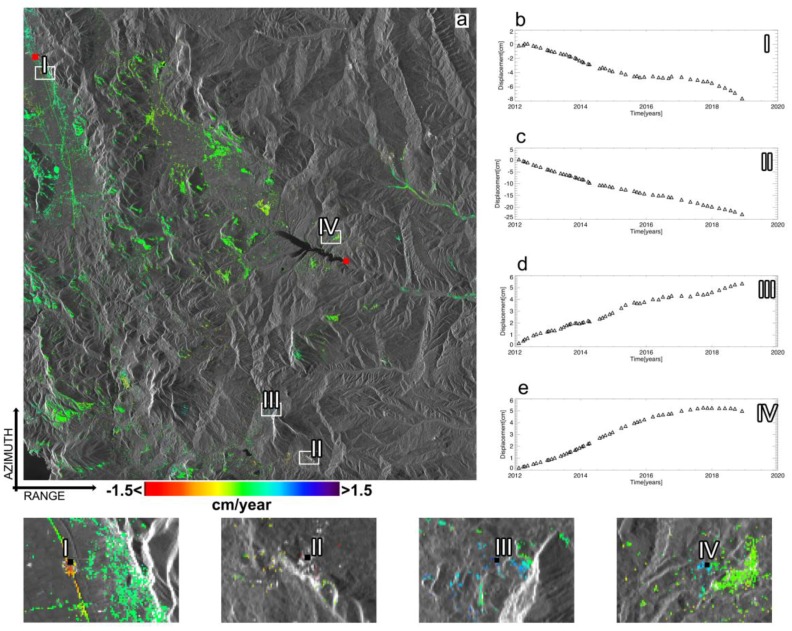
Un-weighted SBAS approach InSAR products. (**a**) Mean displacement velocity map of the investigated area. Four SAR pixels, labelled as I, II, III, and IV have been selected. The relevant ground displacement time-series are shown in (**b**–**e**), whereas the zoomed views of the areas nearby the selected pixels are also shown.

**Figure 10 sensors-20-01103-f010:**
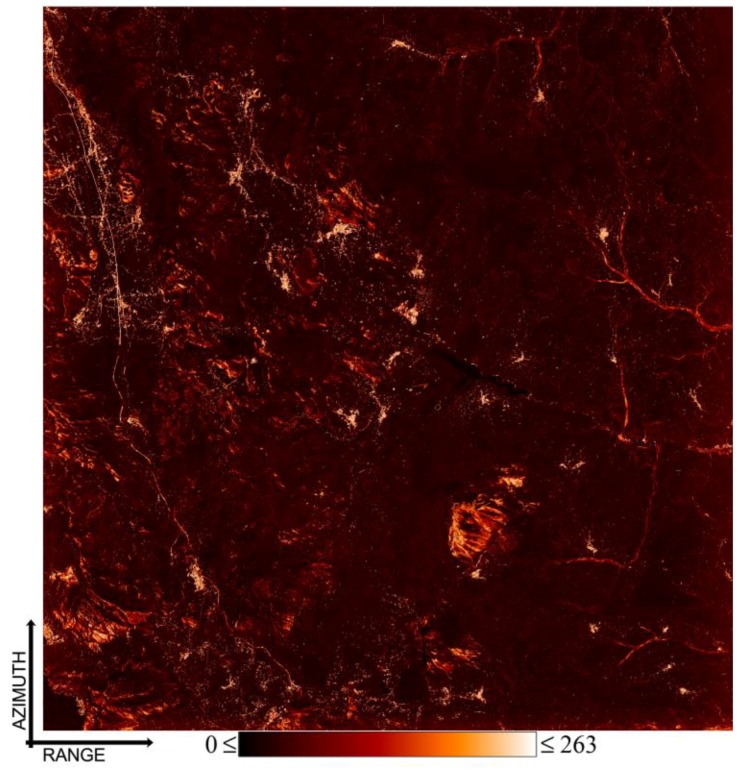
A spatial map of the investigated area in SAR coordinates related to the number of interferograms used for each pixel by the WSVD time-series inversion.

**Figure 11 sensors-20-01103-f011:**
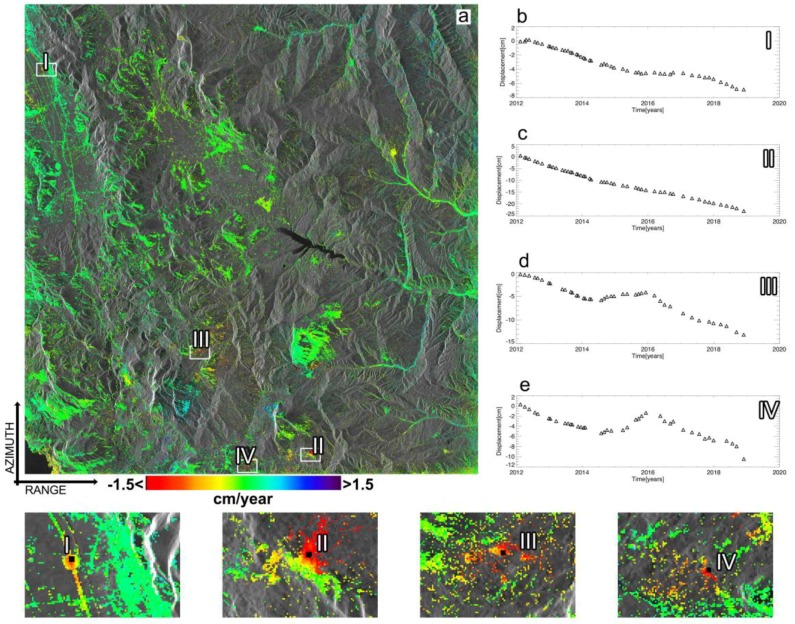
WAVE approach InSAR products. (**a**) Mean displacement velocity map of the investigated area. Four SAR pixels, labelled as I, II, III, and IV have been selected. The relevant ground displacement time-series are shown in (**b**–**e**), whereas the zoomed views of the areas nearby the selected pixels are also shown. Note that the time-series of pixel III and IV are characterized by 42 and 40 acquisition times, respectively.

**Figure 12 sensors-20-01103-f012:**
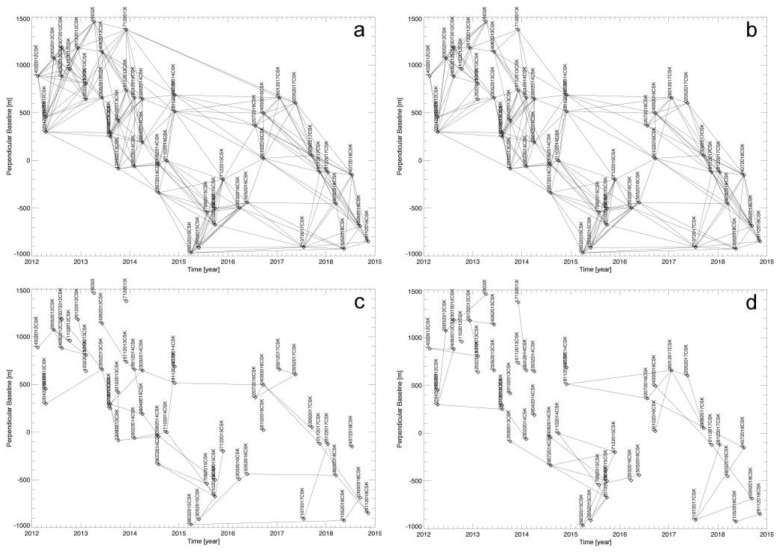
(**a**)–(**d**) InSAR distribution in the temporal/perpendicular baseline plane related to the SAR pixels labelled as I, II, III, and IV in Figure 11.

**Figure 13 sensors-20-01103-f013:**
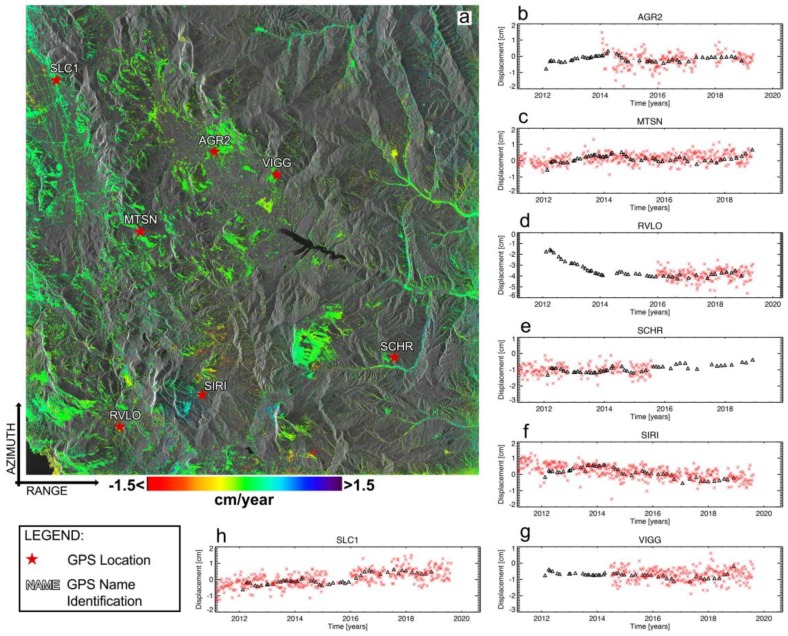
Comparison between the GPS measurements and the LOS-projected deformation time-series in correspondence with the seven available GPS stations of the area. (**a**) Mean displacement velocity map of the area. (**b**–**h**) Plots of the LOS-projected ground deformation time-series (black triangles) and the GPS deformation measurements projected along the radar sensor LOS (red crosses).

**Table 1 sensors-20-01103-t001:** The Used COSMO-SkyMed Dataset.

Day	Month	Year	Orthogonal Baseline [m]	Doppler Centroid [Hz]
14	02	2012	887.96	−1164.2
02	04	2012	300.52	−1136.8
18	04	2012	455.29	−1074.5
20	05	2012	1072.3	−1241.8
23	07	2012	1186.6	−1133.3
24	08	2012	878.17	−1176.1
11	10	2012	959.50	−1235.1
26	12	2012	1177.6	−653.37
11	01	2013	811.07	−659.16
12	02	2013	642.53	−681.40
16	03	2013	1458.4	−536.96
23	05	2013	660.91	−1092.4
24	06	2013	1139.4	−1176.9
26	07	2013	249.04	−1188.3
27	08	2013	291.91	−1201.2
12	09	2013	−83.779	−1194.1
30	10	2013	416.50	−1196.9
15	11	2013	739.37	−1187.5
17	12	2013	1373.2	−1231.9
18	01	2014	656.69	−1237.0
03	02	2014	−62.523	−1206.4
23	03	2014	646.41	−1163.4
08	04	2014	191.51	−1099.6
29	07	2014	−337.35	−1227.3
30	08	2014	−42.907	−1206.0
01 *_M_*	10 *_M_*	2014 *_M_*	0.0000 *_M_*	−1236.5 *_M_*
18	11	2014	517.28	−1194.4
20	12	2014	685.74	−1211.9
26	03	2015	−967.52	−1224.9
13	05	2015	−912.84	−1182.2
17	08	2015	−540.35	−1245.1
18	09	2015	−497.50	−1325.8
20	10	2015	−672.71	−1384.0
07	12	2015	−198.14	−1189.8
12	03	2016	−494.75	−1284.2
15	05	2016	−437.48	−1081.5
02	07	2016	366.08	−1185.4
04	09	2016	498.12	−1258.2
06	10	2016	24.905	−1266.8
26	01	2017	659.14	−1252.1
02	05	2017	605.27	−1209.8
21	07	2017	−909.00	−1334.3
23	09	2017	52.237	−1414.1
10	11	2017	−119.79	−1291.0
28	12	2017	−123.48	−1521.1
18	03	2018	−450.21	−1258.0
21	05	2018	−924.02	−1275.6
24	07	2018	−153.99	−1218.1
10	09	2018	−682.59	−1296.5
29	11	2018	−848.53	−1290.5

*^M^* The footer corresponds to the master acquisition used for the images registration.

**Table 2 sensors-20-01103-t002:** Root Mean Square Error (RMSE) of the difference between SAR and GPS LOS-projected measurements.

Figure Identifier	Name	RMSE [mm]
13.b	AGR2	3.75
13.c	MTSN	3.19
13.d	RVLO	3.34
13.e	SCHR	3.17
13.f	SIRI	2.82
13.g	VIGG	3.10
13.h	SLC1	3.49
Mean RMSE		3.27

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
