# Peer review of "On the Use of Weighted Least-Squares Approaches for Differential Interferometric SAR Analyses: The Weighted Adaptive Variable-lEngth (WAVE) Technique"

_sensors, 2020, doi:10.3390/s20041103_

Round 1
Reviewer 1 Report
The paper proposes a weighted adaptive variable length algorithm for SBAS InSAR processing, which makes good use of the small baseline and high coherent interferograms in Phase unwrapping and WLS inversion. The algorithm first uses the adaptation EMCF PU method, and use the two step network with small baseline and redundant interferograms, which may improve the phase unwrapping quality of low coherent regions; then the algorithm uses the WLS method to estimate deformation results after selecting the good interferograms and removing the isolated interferograms. The idea is very interesting, but there are still some points which are not clear:
1.In Section 2.2, the author proposes the two step EMCF PU method by complementing additional interferogrmas. But Fig 2 is not very clear, the author should give in details about the redundant baseline network.
2.In Section 2.3, the paper uses the simplified weight function for WLS inversion, with the assumption that wrapped phase obeys norm distribution. But the weight is used for unwrapped phase, which seems to be unreasonable due to the different statistics distribution for the unwrapped phase.
3. In Experiment, the paper only gives the deformation results by WLS method, and we suggest the author would provide the comparison results between unweighted and weighted methods, to better certify the advantages of weighted method.
4. The introduction to SBAS InSAR technique is partial, we recommend the author add more reference, like STAMPS method and Mintpy method.
Author Response
Please, see the attached document.

Reviewer 2 Report
The article is well describing a novel technique bringing and additional value of coherence-based weighting to a SBAS-based processing chain.
The article is ready to publish as it is and I see only few minor improvements that could be introduced, e.g. word matrixes --> matrices (e.g. line 465)
What I miss in the article is some critical study or at least discussion on possible negative aspects of introducing the weighting. If authors are to provide an update on the paper, it would be great if they could focus few sentences on discussing the effect for cases where one interferogram of a significantly higher weight (high coherence) would be actually not due to a deformation - in such case, in my opinion, a bias may be introduced that would normally be not significant in case of unweighted approach.
However I recommend the article to be published in the current state.
Author Response
See the attached file, please.

Reviewer 3 Report
This paper concentrates on the study of the weighted Least-squares (WLS) approaches for 12 the generation of ground displacement time-series through differential interferometric SAR 13 (DInSAR) methods. I have a few comments for the purpose of the improvement:
The Figure 4 is not very clear, the author should give in details. As Section 2.3, It seems to be impossile. Please confirm what will you do next.Author Response
Please, see the attached file.

Round 2
Reviewer 1 Report
Here are a few of my doubts, and I hope to get a further explanation to make them clearly,
1) it is not clear about how to select the coherent pixles for deformation study. Is it selected on a pixel-by-pixel basis by considering the coherence of each pixel in every inSAR pairs?
2) There are different inSAR pairs for each pixles just like figue 4 shown? So is the figue 1 the same as the figue 4, which is relate to a distinctive pixel? Or the InSAR pairs in figure 4 for each pixels are included in figure 1?it is not clear in the paper.
3) There are many weigted LS methods used in MT-InSAR, such as BFGS in PTA(SqueeSAR), Weight-Coherence Phase Link(WCPL), what are the advantages over these methods?
4) the results obtained by WAVE is campared with the un-weighted SBAS approach in this study, how about the result campared with the SqueeSAR, considering the total selected pixel number/density, the computing efficiency?
Author Response
see the attached document
